# Exploring the Potentials of Artificial Intelligence Image Generators for Educating the History of Architecture

Mohamed W. Fareed [1], Ali Bou Nassif [2] and Eslam Nofal [1,3,*]

1 Department of Architectural Engineering, University of Sharjah, Sharjah 27272, United Arab Emirates; u22106700@sharjah.ac.ae
2 Department of Computer Engineering, University of Sharjah, Sharjah 27272, United Arab Emirates; anassif@sharjah.ac.ae
3 Department of Architectural Engineering, Assiut University, Assiut 71516, Egypt
* Correspondence: enofal@sharjah.ac.ae

**Abstract:** The rapid integration of Artificial Intelligence (AI) tools, specifically text-to-image generators, across various domains has had a profound impact on numerous fields. Despite this, the potential applications of AI image generators in architectural education, particularly in teaching the history of architecture, remain underexplored. This research aims to uncover the possibilities of utilizing AI image generators, with a specific focus on the capabilities of Leonardo AI, to enhance communication and engagement. This study employed an experimental methodology to investigate how the integration of AI image generators in education on the subject of "History of Architecture" promises to elevate the learning experience, offering new perspectives, visualizations, and interactive tools. Two workshops were conducted with university students to explore AI image generators' potential applications in architectural history education. The first workshop utilized an iterative approach, while the second aimed to assess students' analytical skills. The ultimate objective was to determine the capabilities of this tool and stimulate discussions regarding its potential future implementations. Following the workshops, online questionnaires were administered to students, and interviews were conducted with educators. The findings of this research underscore the need for validating AI-generated images, establishing guidelines to prevent misuse, and designing tailored AI tools for History of Architecture courses, thereby paving the way for further advancements in architectural history education.

**Keywords:** AI image generation; artificial intelligence; architectural education; history of architecture; Leonardo AI



## 1. Introduction

### 1.1. Context

The evolution of architecture is linked to the social and historical development of society, as architecture is a social discipline. In order to prepare future architects for a variety of duties, including those that are frequently connected to the study of social history, or the history and philosophy of architecture and art, architecture schools incorporate courses on the history of architecture in the academic curriculum. Architectural history is important in building the understanding of future architects. It offers opportunities to interpret various aspects such as form, spatial analysis, function, and development trajectories related to built environment and their historical background, including social, economic, and technological factors specific to each era and location [1–3]. Traditionally, within the course syllabus, educators commonly use tools such as photographs, maps, drawings, physical models, and occasional field trips to convey the three-dimensional spatial characteristics of a particular building. As a supporting approach, some educational institutions incorporate alternative digital tools into their teaching methodologies, providing students with a more comprehensive and practical learning experience regarding a particular spatial context [4].

The fast-rising evolution of Artificial Intelligence (AI) tools, including the emergence of text-to-image generators, presents a unique challenge for architectural schools [5], particularly in the context of theoretical courses such as "History of Architecture". The traditional dependency on classic textbooks and conventional classroom lectures for such classes has caused difficulties in adapting to the digital age's dynamic landscape. Student engagement has become a crucial factor in the learning process, and conventional teaching methods risk losing appeal, especially with smartphones as potential distraction sources [6]. Recognizing this need for innovation and addressing the limitations of current teaching methods, integrating AI image generators into the field of architectural history holds significant potential for capturing students' attention and bridging the gap between theoretical comprehension and practical engagement, offering a forward-thinking perspective. AI can contribute to visualizing historical spaces, personalizing learning paths, fostering collaborative learning platforms, creating a more nuanced understanding of architectural styles over time, and enabling simulation and scenario building.

In this sense, an educational concept such as the Zone of Proximal Development (ZPD) [7–9] has great significance. When applied judiciously, it can guide students beyond their current cognitive grasp, navigating the balance between challenge and mental aspects in the domain of architectural history. Characterized by the exchange of complex design principles and the evolution of styles, the ZPD emerges as a suitable framework.

This research contends that AI image generators present a revolutionary pedagogical tool for reshaping the landscape of architectural education by offering visual depictions of architectural concepts and historic structures; these generators ignite students to go beyond their cognitive thresholds, fostering a nuanced understanding of architectural history, as depicted in Figure 1. Considering this, it is vital to note that, while integrating AI into architectural education, particularly the HoA, may be beneficial in certain aspects, it may also bring challenges. It should not be intended to entirely replace existing teaching methods. Therefore, educators are tasked with preparing students for a future where AI plays a central role in architectural practices, emphasizing the importance of human insight and creativity in architectural education [10].

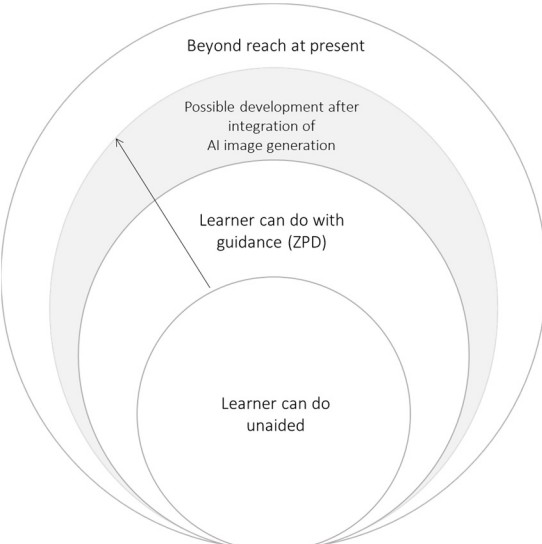

**Figure 1.** The proposed integration of AI image generation represented using the zone of proximal development framework.

### 1.2. Research Problem

While AI image generators have gained popularity in recent years, their practical implementation in teaching the history of architecture still needs to be improved, and more literature is required to discuss the pros and cons of these tools in this domain. In this research, it is argued that using AI tools might boost performance in learning and

create an active, interactive pedagogic atmosphere by applying Bloom's theory, which is a hierarchical framework used to classify educational objectives into levels of cognitive complexity. Originally developed by educational psychologist Benjamin Bloom in the 1950s, the taxonomy categorizes learning into six levels: Remembering, Understanding, Applying, Analyzing, Evaluating, and Creating. These levels represent progressively more complex cognitive processes, from the simple recall of information to the synthesis of new ideas [11]. Applying Bloom's taxonomy to the History of Architecture course reveals a field predominantly dependent on conventional tools but incorporating evolving technologies such as Virtual Reality (VR), Augmented Reality (AR), and others (Figure 2).

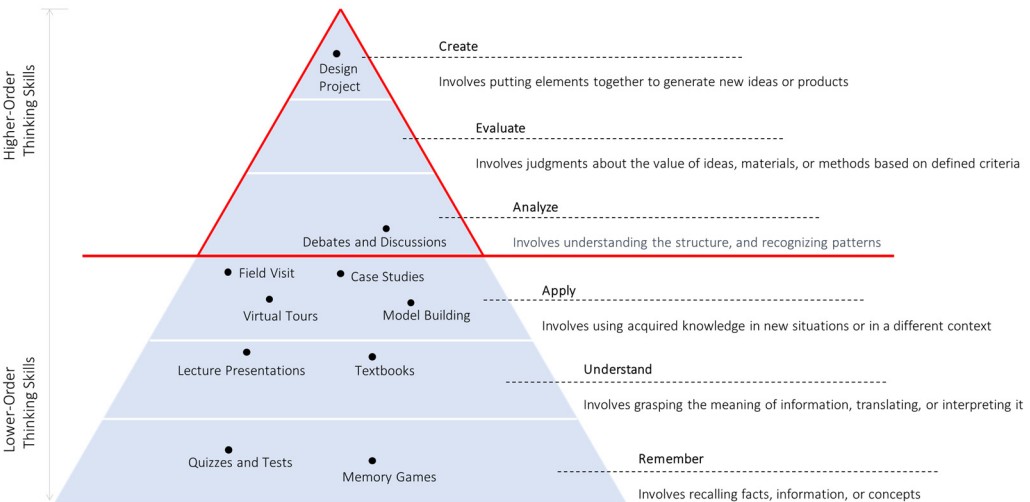

**Figure 2.** The current tools ranked using Bloom's theory.

For "Remember", at the first level, traditional tools such as textbooks, formal lectures, and static images are usually used in History of Architecture courses, which include basic facts about architectural styles, historical periods, and landmarks.

When shifting to the "Understand" and "Apply" levels; these tools enable one to comprehend architectural principles or factors influencing previous designs. Mainly, this includes using a case study, a site visit, or assignments in which students apply the skills they have obtained to deal with particular architectural issues. Figure 2 shows the gap while transitioning from low- to high-order thinking skills, as most current tools are of a lower-order ranking.

The limitations become apparent as one goes up in the hierarchy, passing through "analysis", "evaluation", and "creation". Despite the gains that current digital tools such as VR and AR have brought toward improving how historical spaces are experienced, these technologies target aspects such as the "application" and "analysis". They permit the students to investigate the structures. However, they cannot effectively stimulate the integration of different scenarios of architectural theories and the study of their historical relevance. In contrast to static images or scripted VR scenarios, AI image generators have the ability to adjust to students' learning requirements by creating an individualized and progressive learning process that follows Bloom's taxonomy.

Further, AI image generators may present a distinct competence at the "evaluation" stage of student analysis related to architectural developments' historical relevance and social influence. The visual representation of architecture may allow students to view the impact these buildings/spaces had at various times for specific societies.

While endowed with VR, AR, and digital devices, no overall approach integrates well with Bloom's taxonomy at the highest-order thinking levels denoted by creation. This gap can be bridged by coupling AI image generators with human intelligence to enhance the students' engagement throughout HoA courses.

## 2. Related Work

### 2.1. Why and How to Teach the History of Architecture?

Architectural history is essential for understanding architecture, culture, and development perceptions. The significance of delving into the History of Architecture field lies in its role as applied history in understanding and drawing lessons for addressing contemporary and future challenges. Students explore forces such as social, economic, political, climatic, and technological factors, as well as resource availability, across diverse spatial and temporal scales that have shaped the built environment. The core element of architectural history learning is how different eras used architecture as an art form. This is presented through interweaving a multi-layered fabric that combines art history, archaeology, and historiography. The academic domain teaches different architectural styles, methods, and materials analysis. In most cases, this is accomplished by thoroughly analyzing specific architectural wonders and putting them in their cultural settings [12,13]. A well-correlated integration of Design Studio courses, core courses, and practice can benefit the built environment's future [14].

Collectively referred to as Digital Reality, Augmented Reality (AR), Virtual Reality (VR), Mixed Reality (MR), and Artificial Intelligence (AI), these technologies have improved cognitive abilities, including abstract reasoning, spatial recognition, visible short-term reminiscence, and multi-tasking capabilities. However, with this alteration, the limits between traditional coaching techniques and virtual gear must be addressed and delineated. The virtual technology world, immersed in synthetic intelligence, no longer only complements the capability of both student and educator to assimilate and provide knowledge, respectively, but also helps create a symbiotic court between the two [15]. The use of technology in classrooms has been changing through the years. With the cutting-edge era of students more inclined toward visible learning, and new advances in technology like MR, AR, and VR, these technologies are gaining more excellent traction with the cutting-edge generation of beginners, in conjunction with video games. Most of the populace of current-generation college students depend on different varieties of learning and teaching; very few are inclined to spend time reading and examining the facts. Instead, they may be looking at videos and online learning materials, and are more comfortable doing so. College students express an affinity toward other shapes of coaching instead of the conventional fashion of teaching; instructors however express their issues with this visual-based style of teaching [16].

In the past two decades, architectural history education has been affected by the rise of the digital age, which provided an unprecedented opportunity for a reflective assessment of historical incidents and buildings, increased interaction, and information sharing. The presence of modern communication languages and dissemination tools has made it easier for recorded knowledge to be accessed through exhibitions and digital devices. Explanation panels and classroom boards were replaced by innovative methods of interacting with historical content, such as touch-screen monitors, immersive multimedia projection, and video maps of three-dimensional models. Based on their training, architectural historians can expand historical knowledge for authenticated and academic content accessible via the Internet [17].

In a study dealing with the digitization of architectural history [18], the authors underscored the advancements and advantages of computer-aided instruction in the field, emphasizing interpretative instructional techniques over depictive treatments. This approach incorporates experiential tools like digital videos, three-dimensional simulations, and animations to engage students in actively exploring and comprehending course content. Similarly, a study based at Oxford Brookes University [19] demonstrated the importance of combining traditional with digital tools. The researchers determined that a task involving scale-model making and its digital twin, followed by essay writing and formal oral presentations, enhanced students' understanding of the relationship between a building's design, the architect's philosophy, critical analysis, and contextual awareness.

Several scholars have shown interest in recent initiatives incorporating the multi-active participation of students' techniques in history of architecture classes, using IT and pro-

grams in and out of classrooms to enhance the performance results of architecture learners, stating that VR [20,21], AR [22,23], films [24], and similar learning tools significantly boost students' comprehension and motivate them. Once students see a structure's realistic sizes and shapes and look at it in terms of light, shadow, texture, color, and other factors, their understanding is much more significant [25].

*2.2. Text-to-Image Generation in the Architectural Domain*

Text-to-image generation refers to machine methods that are able to convert human-written textual descriptions, such as keywords or phrases, into "visually stunning images" with the same semantic meaning as the text, based on Generative models that can be defined as algorithms that are capable of generating new material/samples based on similar data from previously seen samples from their training dataset. This deep-learning process is carried out utilizing neural networks specially designed and trained to generate samples from training data [26,27]. The generative models are currently applied in many natural language processing (NLP) tools such as text-to-image generation, computer vision, and image recognition. Diffusion models are used for several tasks such as inpainting, image denoising and image generation [28].

The most common types of generative models are: Autoencoders (AE), Generative Adversarial Networks (GAN), Deep Boltzmann machines (DBMs), and Diffusion models. According to Dhariwal and Nichol [29], diffusion modeling is a cutting-edge technique for generating images efficiently. This method involves progressively adding noise to an initial image without revisiting previous iterations. Each step in the process starts anew, independent of past steps, resulting in a final image consisting solely of random noise. In this process, the concept of conditioning plays a crucial role. Conditioning involves guiding the noise predictor to achieve a desired image outcome by subtracting the predicted noise. To make this concept understandable to machines, text must first be tokenized, converting words into numerical representations. The tokenization process is facilitated by a CLIP tokenizer, which is short for Contrastive Language-Image Pre-training. This approach leverages both text and image data to train natural-language-processing tasks, such as language understanding, text classification, and translation. By training a model to predict the correspondence between text and images, CLIP improves the representation of textual data by incorporating visual information. Following tokenization, embedding is employed to represent words or tokens as continuous, low-dimensional vectors. The goal of embedding is to capture the semantic and contextual meaning of words or tokens, facilitating their integration into other models, including neural networks [30].

Text-to-image generation involves transforming human-readable textual descriptions, including phrases and keywords, into visually appealing and meaningfully related images. Initially, approaches used in image synthesis research were based on text-matching methods involving supervising techniques to link words and graphics. The prompt engineering process requires naming specific object(s), comprehending natural language, and understanding common-sense knowledge to formulate sensible and relevant descriptions. Thus, the AI tool understands significant aspects of an image, realizes their interconnection, and builds content accordingly [31,32].

Using AI-generated images will open new ways of looking at classic architectural styles using modern techniques or imagining them as future products. In a recent design study by [33], fifteen visualizations were created using Midjourney[1]. These visualizations present different interpretations of renowned architectural models that existed in various architectural styles during different periods, as if other architects had designed them. In [34], a series of baroque façades was reimagined in a surreal contemporary fashion. Likewise, [35] depicted the famous Versailles palace through opulent and ornate styles. Another example is given by [36], which shows an ancient architectural type visualized in a modern form inspired by ancient Mesopotamian Ziggurats, illustrating how historical components can coexist with contemporality, overcoming cultural barriers, and celebrating local cultures. Similarly, [37] was influenced by abstract expressionism and used it to

capture the grandeur of ancient Egyptian architecture. The "Post Pharaonic Architecture" series displays immense stone structures and huge monolithic blocks.

Generative art has been a research topic for years, but only found public interest with the advent of diffusion model platforms such as DALL-E[2], Midjourney, Leonardo AI[3], or Stable Diffusion[4]. These models depend heavily on an internal model architecture that applies the "interface" paradigm. Although they undertake similar core workflows, their distinctions stem from their unique interface approaches. Notably, they involve the progressive resizing of a higher resolution and the comparing of many image variations before sampling the best versions via different means. Unlike DALL-E and Stable Diffusion, editing an uploaded image or a previous result directly using img2img-based operations on inpainting and outpainting is possible.

Notably, image generation across all models takes only a few tens of seconds, making them suitable for quick, creative sessions alone or with clients. They also facilitate the import of external images, enabling composite workflows. However, the training data details could be improved, with Stable Diffusion being known to use the LAION-5B dataset, sourced from web-scraped image and text data. While the specific training data for DALL-E and Midjourney could be more precise, these models have distinct image styles. DALL-E and Stable Diffusion are more effective in both drawn and photorealistic outputs. At the same time, Midjourney tends towards a more artistic style, especially in earlier versions [38].

A comparative evaluation of the most widespread AI image generators in Table 1 shows that Midjourney exhibits features such as developing qualitative pictures with fin elaborations and lifelike textures. At the same time, Stable Diffusion Models generate regular images that satisfy high-fidelity or discolored realism. Leonardo, to suit variants inspired by DALL-E, features creative image synthesis and customizations. Concerning the data used to train the models, it is known that DALL-E uses a large and varied dataset. However, specific details on the datasets for Midjourney, Stable Diffusion, and Leonardo may also differ. Image generation types may vary across the models, since DALL-E and Leonardo are the best creative and customizable image synthesis performers. User-friendliness depends on the interface and implementation, with every model catering to different user preferences. Pricing models for these AI image generators could differ from open-source availability to commercial licenses and, therefore, require consideration based on individual needs and usage requirements. Leonardo is the cheapest and easiest tool to date, and therefore it was chosen to be used during the conducted workshops.

**Table 1.** Comparison of the most widespread AI image generators.

| Criteria | DALL-E | Midjourney | Stable Diffusion | Leonardo AI |
|---|---|---|---|---|
| Training Data | Diverse datasets | Custom datasets | Custom datasets | Diverse datasets |
| Image Generation Type | General-purpose | Custom illustrations | Custom illustrations | General-purpose |
| Creative Capabilities | Diverse and creative | Artistic illustrations | Artistic illustrations | Diverse and creative |
| User-Friendliness | Requires tech skills | User-friendly | User-friendly | User-friendly |
| Pricing Model | Free for public use | Subscription-based | Subscription-based | Partially free |

## 3. Methodology

Two experimental one-day workshops were conducted to investigate AI image generators' potential applications in architectural history education. The workshops took place under the supervision of the authors at the Department of Architectural Engineering, the University of Sharjah, UAE. The learning scenarios for the workshops were scheduled as an optional section/extra activity within the current curriculum of the HoA course. All the students enrolled in the course were invited to participate, with no pre-experiences required. The HoA course offers a comprehensive study of cultural heritage preservation and management. It begins with an exploration of major ancient civilizations, analyzing the impact of aesthetic and symbolic elements on their architectural forms, and comparing architectural styles across civilizations. Both workshops were designed to accommodate

a range of 10 to 15 students enrolled in HoA courses, and were conducted online via the Microsoft Teams platform. Educational materials such as presentation slides, examples of past uses, and breakout rooms for collaborative group discussions were integrated. Each workshop was scheduled for 100 to 120 min, including a short break. The structure was organized into several sessions encompassing introductory segments, following sessions aimed at comprehending text-to-image tools, hands-on exploration of the operational aspects of these tools (with a particular focus on the AI image-generation tool named "Leonardo AI"), and group discussions designed to facilitate the exchange of ideas and insights, as well as the exploration of potential future directions within this domain. Following the conclusion of these workshops, comprehensive feedback and comments were gathered from educators and students. This evaluative process was carried out through online Google Forms, providing us with the data needed to develop discussion points, future research directions, and conclusions.

In the context of this research, Bloom's taxonomy was used as the main theoretical foundation model of the research for several reasons:

- Bloom's taxonomy was used to justify the research problem of lacking higher-order thinking activities in the current curricula of HoA. In Bloom's model, more focus is given to the development of these thinking skills, which aligns with the multiple nature of architectural history and is essential in the formation of future architects in both scholarship and practice.
- Recent literature reviews have proved that Bloom's model is effective when dealing with history of architecture courses. It can be used to measure and assess different teaching methods in order to develop better comprehension, understanding, and analytical ability [18]. Moreover, the model can also be utilized in adapting new potential technologies such as Virtual Reality (VR) [39] and video-based learning resources [11] to existing courses on architectural history, aiming to maximize the potential of pedagogical strategies to improve student learning outcomes.
- In line with the learning strategy of the University of Sharjah, Bloom's model provides a structured framework for learning objectives, required skills, and competencies that students need to acquire during the course. Moreover, Bloom's model is not built specifically for any particular subject, making the idea of this research suitable to be generalized, replicated and re-adapted by future interdisciplinary applications. Research on AI-generated images is still in its very early stages; hence other domains such as art history, cultural studies, and the visual arts can benefit and improve this research idea.
- Unlike other approaches that are more specialized or theoretical in nature, Bloom's taxonomy offers a user-friendly framework and a common language that can enhance engagement and accessibility for learners of diverse backgrounds and abilities across different educational settings.

Below are the research hypotheses that were developed in this direction in accordance with the research's purpose and chosen research methodology.

**H1:** *The validity and reliability of AI-generated images need to be rigorously checked by both educators and students.*

**H2:** *AI-generated images can be a tool for educating non-experts.*

**H3:** *AI image generators are able to recognize different historic architectural styles.*

**H4:** *AI-generated images need to be coupled with human intelligence for a better educational experience.*

**H5:** *AI image generators can help the visualization of alternative histories of unfinished or damaged architecture, not only in the physical aspect but also in the virtual realm.*

### 3.1. Learning Scenario for Workshop A

The first workshop, conducted in a hybrid format on 13 September 2023, targeted MSc students enrolled in the HoA remedial course as part of the Master's Program in the Conservation Management of Cultural Heritage. This program, a collaboration between the University of Sharjah and the ICCROM regional office in Sharjah, United Arab Emirates [40], aimed to serve heritage professionals from diverse backgrounds in the Arab region. The workshop was primarily conducted in Arabic, aligning with the program's official language.

The workshop aimed to enhance the educational experience in the HoA course through text-to-image tools. Participants explored the intersection of AI and image-generation technology to visually represent historical architectural landmarks based on textual descriptions using an iterative approach [41]. Participants were encouraged to familiarize themselves with text-to-image generation tools like Leonardo or Midjourney to prepare for the workshop.

The workshop consisted of six sessions: Session 1 began with a welcome and introductions, followed by an overview of the objectives and the structure. It discussed the definition, importance, and relevance of these tools to architectural history, with case studies highlighting successful applications. Session 2 involved a hands-on exploration, introducing various text-to-image tools and practical demonstrations of generating architectural imagery from textual descriptions. In Session 3, the group tasks were presented, while Session 4 involved group discussions, with breakout groups analyzing specific textbook sections corresponding to architectural periods covered in the course. Each group formulated prompts to visualize a particular building from an ancient era studied during the procedure. Session 5 included sharing ideas and insights, wherein each group presented potential applications of text-to-image tools based on their discussions. The last session concluded with the exploration of future directions.

### 3.2. Learning Scenario for Workshop B

The second online workshop, conducted entirely in English on 23 September 2023, targeted undergraduate students majoring in architectural engineering at the University of Sharjah, UAE [42]. This session started with a short introduction to AI, followed by students being assigned to form prompts for four main tasks based on insights, ideas, concepts, and experiences gained from previous traditional lectures in the "Hof A" course.

In the first task, students were asked to visualize a building combining features from two or more different historical architectural styles, such as ancient Persian and Egyptian. The objective was to ignite contextual thinking and creativity by encouraging the exploration that can be produced from such architectural fusion.

In the second task, students were tasked with visualizing a modern building using architectural laments and styles from a specific historical period. The aim was to connect the importance of history and contemporary design to check how AI tools can interpret heritage ideas.

The third task focused on comparing human creativity with AI abilities. Students had to draw a fast conceptual sketch of a building or architectural element that interested them in recent lectures; they then described what they had already drawn to the AI image generator. They compared the AI-generated image with their drawings. The objective was to facilitate discussions on AI's strengths and limitations in visualizing architecture.

In the final task, students were tasked with reimagining unfinished, damaged, or lost ancient buildings using their knowledge about the history of architecture and AI image generators. The goal was to develop students' historical understanding while testing the accuracy and creativity of AI image generators in architectural reconstruction.

### 3.3. Feedback Questionnaire from Students

The student responses to the introduced tool for Workshop A and Workshop B were measured with qualitative and quantitative questions through online Google Forms.

The Workshop A questionnaire started with general information such as the individual's age, country of origin, and academic qualification. Furthermore, the participants were advised to give subtle feedback on their final rating about their level of satisfaction with the workshop, whether it was clear to them, how understandable the provided tools were, and their use in that sense. In the next part of the workshop questionnaire, the users rated various components on a 5-point Likert scale concerning the usage of AI in design education. It included evaluations such as understanding AI applications in the context. Furthermore, participants explained how the tool helped to explain different parts of architectural history, such as historical notions, functional principles, aesthetic considerations, authentic details, or non-physical qualities.

Meanwhile, the questionnaire tailored for Workshop-B participants adopted a distinct structure: the initial section, labeled "A. General Information", focused on a broader demographic context, seeking information on country of origin, age, familiarity with AI image generators (rated on a 5-point Likert scale), and the number of years dedicated to studying architecture. Section B, designated "B. Mini Experiments", constituted a pivotal component of the Workshop B questionnaire. For each of the four covered tasks, participants provided detailed ratings assessing their confidence in AI's ability, satisfaction with the final products, and challenges encountered during the experiments.

### 3.4. Interview with Educators

Semi-structured video and email interviews were conducted with experts and academics teaching the History of Architecture. The main idea behind this was to present the outcomes of the two workshops and then discuss the possible positive or negative implications of the proposed integration of AI image generators in teaching HoA courses.

The discussions revolved around four main points. The first was about the benefits of employing AI image generators as a pedagogical tool for teaching architectural history. This inquiry aimed to extract insights into such technology's unique contributions and advantages in the educational context.

The second point was to ask the experts to share their considerations and concerns about integrating AI into educating students about architectural history; of particular interest were reflections on ensuring the accuracy and reliability of the AI-generated images, and addressing potential challenges that may arise in this regard.

The third point focused on assessing the effectiveness of AI-generated images in facilitating student learning and engagement within architectural history. Educators were encouraged to share their observations and experiences, shedding light on the impact of these technological tools on student comprehension and involvement in the subject matter. Lastly, the educators were invited to share their perspectives on the future trajectory of AI image generators in architectural history education.

## 4. Results

### 4.1. Results of Workshop A

4.1.1. Observations

Group (1) focused on the ancient city of Ur in Mesopotamia. In (Figure 3a), they started with a basic prompt describing Ur as a walled town. With further experimenting, more complex additions were made. In the prompt (Figure 3b), they added more details on the surrounding landscape and environment, such as "*To build it on top of the hill and above the town*". They added more details to the prompt (Figure 3c) that explored the surrounding contexts related to the city, with its narrow roads and its position beyond the desert. Lastly, in prompt (Figure 3d), the Ziggurat is depicted as "*a three-leveled rectangular building with three stairs leading to-wards a single window on the first level*". In addition, there was a small room above it.

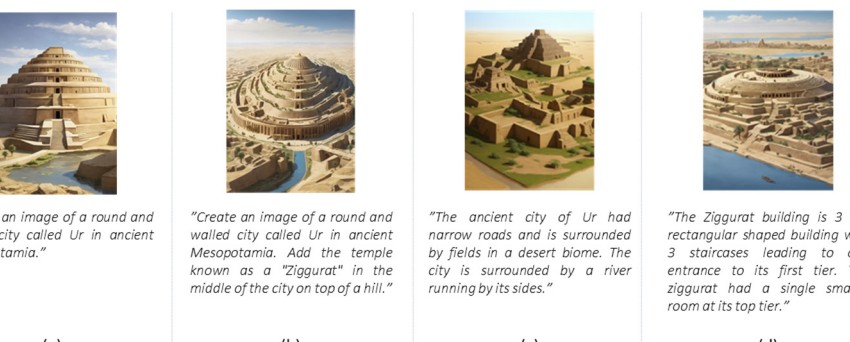

**Figure 3.** In Workshop A, Group (1) focused on the visualization of the Ziggurat as an example from the ancient civilization of Mesopotamia.

The second group's study visualized Çatalhöyük. They started with an initial prompt (Figure 4a), which raised the exciting issue of crowded houses and typical avenues. Later, the prompt (Figure 4b) referred to a galaxy in the night sky showing a new environment and setting to the prehistoric site. The third aspect examined (Figure 4c) further explored the lifestyle and social conditions of those who lived in Çatalhöyük, highlighting the significance of houses in different domains and their daily affairs within their houses. Prompt (Figure 4d) gave more information about the urban form of Çatalhöyük with different shapes and sizes, stating the functions of the middle room and explaining how the dead from other rings were buried under the platforms according to the ritual and religious beliefs of the studied era.

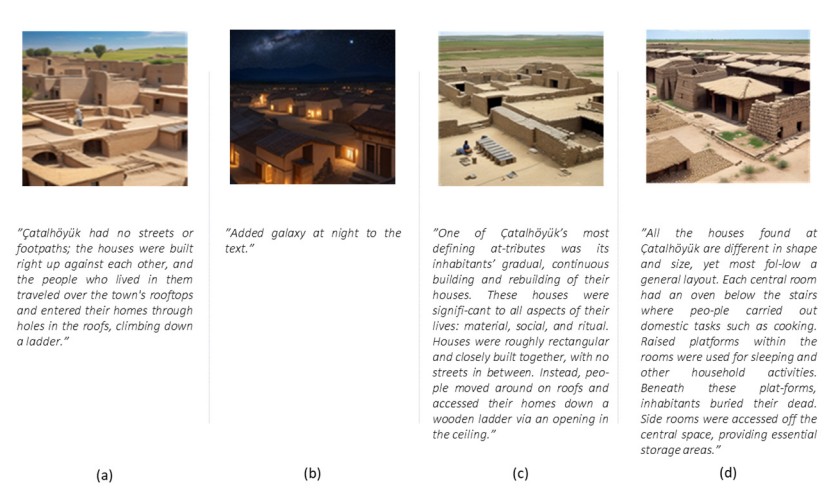

**Figure 4.** In Workshop A, Group (2) focused on the visualization of Çatalhöyük as an example of cities in the pre-historic era.

Students in the third group visualized the archaeological site of the Valley Temple of Khafre, Ancient Egypt, in Figure 5. They prompted the temple's name but found the results inaccurate. Hence, in Figure 5b, the group participants decided to provide additional information. Due to the scarcity of time, they used a chatbot when developing the input prompt, as proposed in [43]. In the second iteration, they added more details regarding the temple's construction, the materials utilized, and the relationship between it and the Great Sphinx. The third iteration (Figure 5c) described building a colossal structure using various kinds of rocks with numerous specifications. Fourthly (Figure 5d), a final try focused on construction materials and stressed the architectural features, particularly the tall, straight, vertical columns with granite blocks.

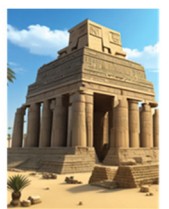 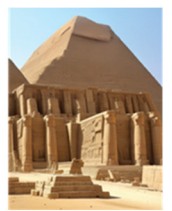 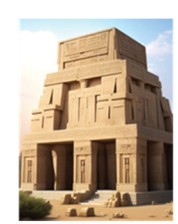 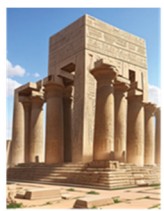

*"Generate a Valley temple of Khafre, built in the ancient Egyptian Dynasty."*

*"Generate a Valley temple of Khafre, built in the ancient Egyptian Dynasty. Khafre› Valley temple, linked to the pyramid by a causeway, was constructed of great monolithic granite blocks and contained remarkable statues of the king carved from diorite. The causeway is located in the Great Sphinx. It was covered using fine limestone partly by ashlars of pink granite. The columns were cubic."*

*"Generate a Valley temple of Khafre, built in the ancient Egyptian Dynasty. The enormous, closed structure with a box-like shape was constructed of great monolithic granite blocks and covered using fine Tura limestone partly by ashlars of pink granite."*

*"Generate a Valley temple of Khafre, built in the ancient Egyptian Dynasty and constructed of great monolithic granite blocks; it was covered using fine limestone partly by ashlars of pink granite. The columns were tall and cubic."*

(a)           (b)           (c)           (d)

**Figure 5.** In Workshop A, Group (3) focused on the visualization of the Valley Temple of Khafre as an example of a temple in ancient Egypt. The prompts were generated by the students with ChatGpt.

### 4.1.2. Feedback from Students

The workshop participants came from different age groups, mostly from their mid-twenties to mid-thirties, and were mainly young UAE professionals in various fields, including visual art, radio, television, the history of Islamic civilization, applied media, tourism studies, tourist guidance, biotechnologies, industrial engineering, and graphic design. Generally, positive feedback (11, *n* = 14) suggested that the AI tool used in the workshop was acceptable, user-friendly, and innovative.

Figure 6 illustrates the results of a questionnaire with 14 participants exploring the potential of AI image generators for educating on the HoA, and several vital insights appear. The participants, numbering 14 in total, expressed notable enthusiasm towards the inventive aspects of AI image generators, with a substantial majority strongly expressing (9, *n* = 14) their potential. Additionally, many participants (7, *n* = 14) strongly agreed that these generators make the educational process easier, while four somewhat agreed, and three remained neutral on this aspect. Furthermore, most participants (8, *n* = 14) expressed strong agreement with the AI image generators' ability to convey historical information, indicating a high perceived efficacy in this educational context. Similarly, a similar consensus was observed regarding the power of AI image generators to convey the meaning of architecture and its artistic and visual values, with eight and ten participants strongly agreeing, respectively. Regarding communicating intangible values, nine participants strongly agreed, three somewhat agreed, and one participant was neutral, while only one participant disagreed. Interestingly, most participants (10, *n* = 14) strongly agreed on the ability to convey realistic and authentic visualization. Finally, when considering the comments provided by participants, a range of sentiments emerged, including positive remarks applauding the potential of AI image generators, some expressing skepticism, and others remaining unsure about their effectiveness in educating on the history of architecture. This diverse set of responses underscores the complex and multifaceted nature of attitudes toward integrating AI technologies into educational practices in architecture.

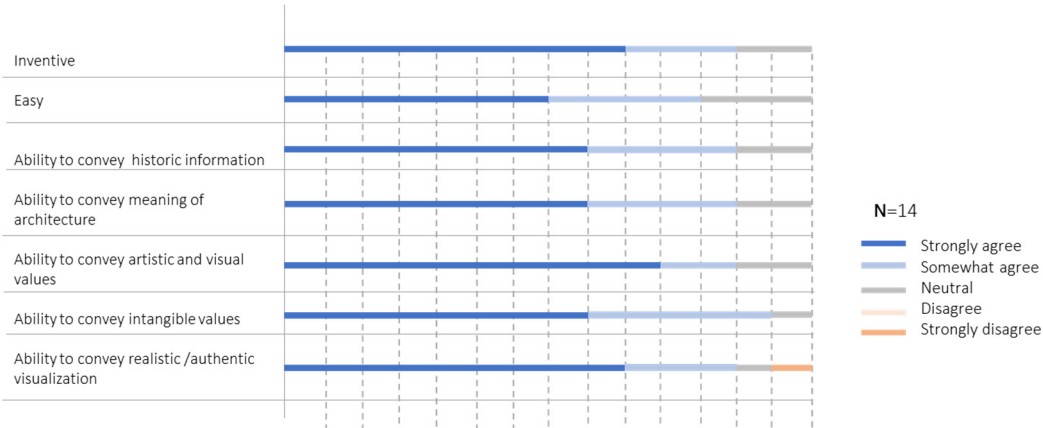

**Figure 6.** Results of the questionnaire (see Appendix A): each row shows an item scaled from strongly agree (blue) to strongly disagree (red), while each cell represents the answer of one participant.

*4.2. Results of Workshop B*

4.2.1. Observations

Students crafted prompts for the four main tasks during the prompt formulation and building selection phase, showcasing their capacity to synthesize insights from prior lectures.

Students integrated features from two different architectural periods in the first task (Figure 7) involving building design with merged historical styles. This hands-on approach underscored their ability to explore architectural concepts beyond conventional boundaries. The merged architectural styles from different historical eras included ancient Persian with ancient Egyptian, Mesopotamian with Persian, Assyrian with Persian, and ancient Egyptian with ancient Greek.

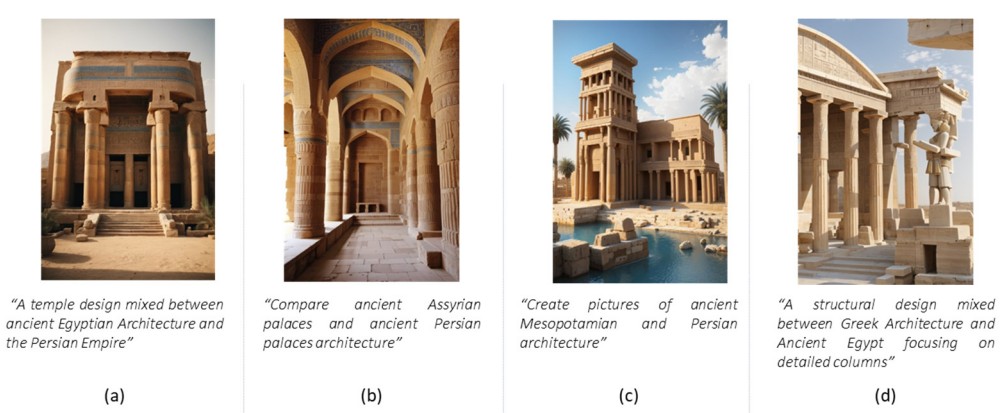

*"A temple design mixed between ancient Egyptian Architecture and the Persian Empire"*

*"Compare ancient Assyrian palaces and ancient Persian palaces architecture"*

*"Create pictures of ancient Mesopotamian and Persian architecture"*

*"A structural design mixed between Greek Architecture and Ancient Egypt focusing on detailed columns"*

(a)                                (b)                                (c)                                (d)

**Figure 7.** In Workshop B, Task 1 focused on combining two different architectural styles.

In Task 2, students depicted modern buildings utilizing architectural details inspired by a specific historical period (Figure 8). This highlighted their understanding of the adaptability of architectural styles and their curiosity about the role of AI in incorporating heritage values into contemporary designs, with examples from various eras, including Brutalism and Gothic periods, and different architectural typologies like high-rise buildings.

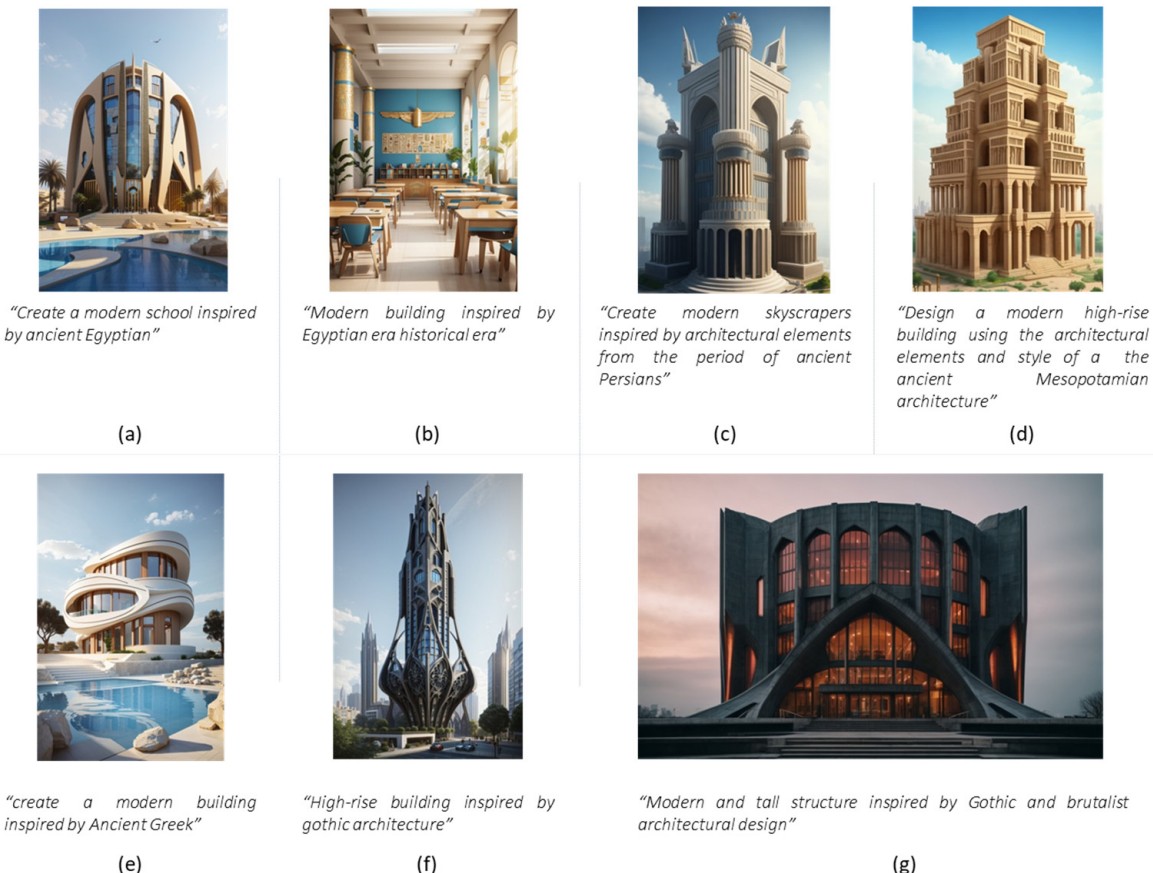

"Create a modern school inspired by ancient Egyptian"

(a)

"Modern building inspired by Egyptian era historical era"

(b)

"Create modern skyscrapers inspired by architectural elements from the period of ancient Persians"

(c)

"Design a modern high-rise building using the architectural elements and style of a the ancient Mesopotamian architecture"

(d)

"create a modern building inspired by Ancient Greek"

(e)

"High-rise building inspired by gothic architecture"

(f)

"Modern and tall structure inspired by Gothic and brutalist architectural design"

(g)

**Figure 8.** In Workshop B, Task 2 focused on visualizing contemporary buildings inspired by ancient styles.

The third task involved prompting students to rapidly sketch specific architectural elements or buildings they had recently studied, followed by providing textual descriptions of their sketches to an AI tool. The hand-drawn sketches (depicted in Figure 9a,b,f) focused on architectural features such as columns and vaults. These sketches showcased the students' analytical understanding of the structures, reflecting their comprehension of layout and three-dimensional aspects. Meanwhile, Figure 9c depicts various architectural typologies, such as the ancient Egyptian Mastaba and the Sumerian Ziggurat. The focus was on assessing the AI's capability to transform these sketches into realistic 3D visuals. The AI-generated counterparts demonstrated an enhanced ability to capture proportions, realism, and colors, effectively translating the students' ideas into tangible representations.

In the last task of revisiting unfinished or damaged historic buildings (Figure 10), students rebuilt ancient structures using their knowledge of architectural history and AI image generators. This exercise tested the accuracy and creativity of AI image generators in architectural reconstruction while showcasing students' historical understanding.

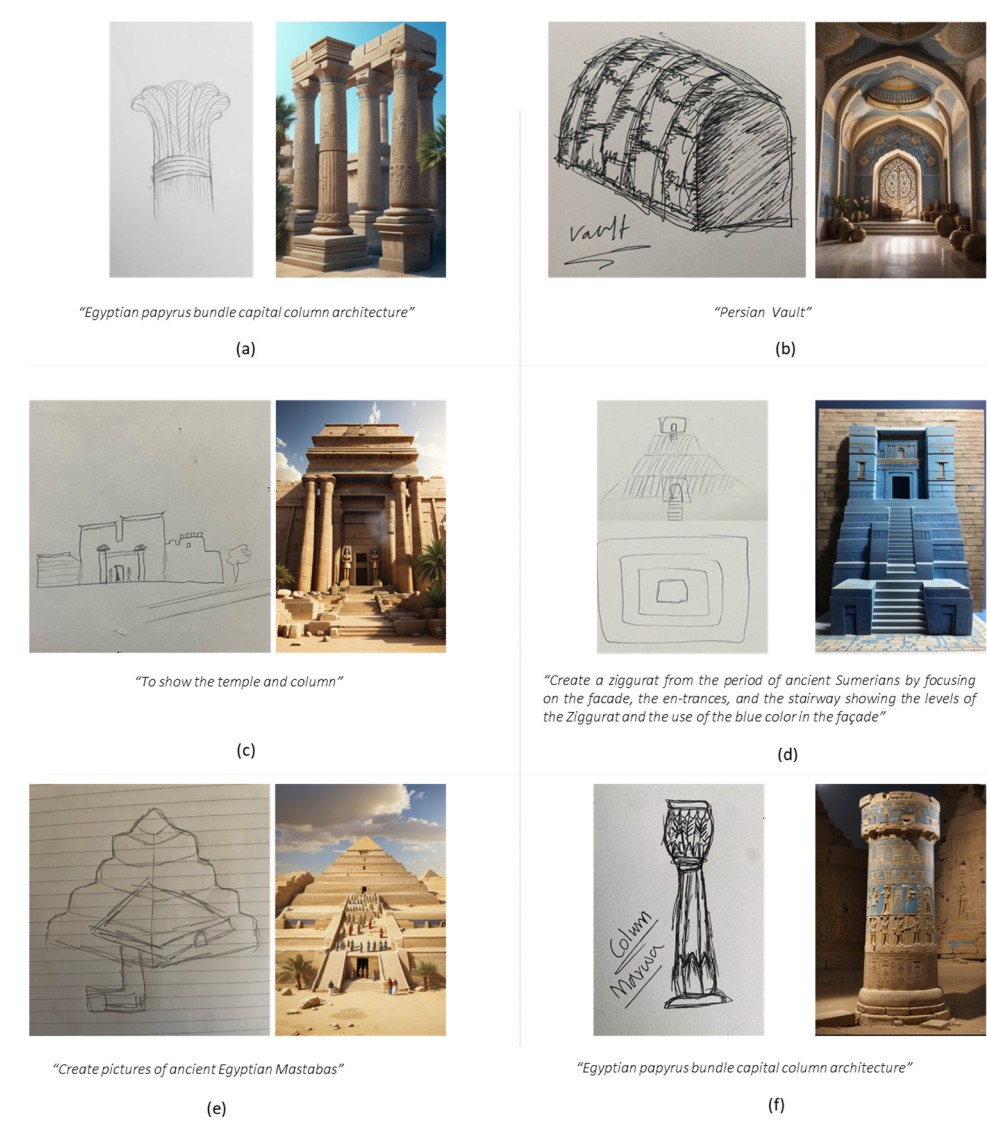

"Egyptian papyrus bundle capital column architecture"

(a)

"Persian Vault"

(b)

"To show the temple and column"

(c)

"Create a ziggurat from the period of ancient Sumerians by focusing on the facade, the en-trances, and the stairway showing the levels of the Ziggurat and the use of the blue color in the façade"

(d)

"Create pictures of ancient Egyptian Mastabas"

(e)

"Egyptian papyrus bundle capital column architecture"

(f)

**Figure 9.** In Workshop B, Task 3 explored the variations between the participants' hand-drawn sketches and the AI-generated images of the same building or element.

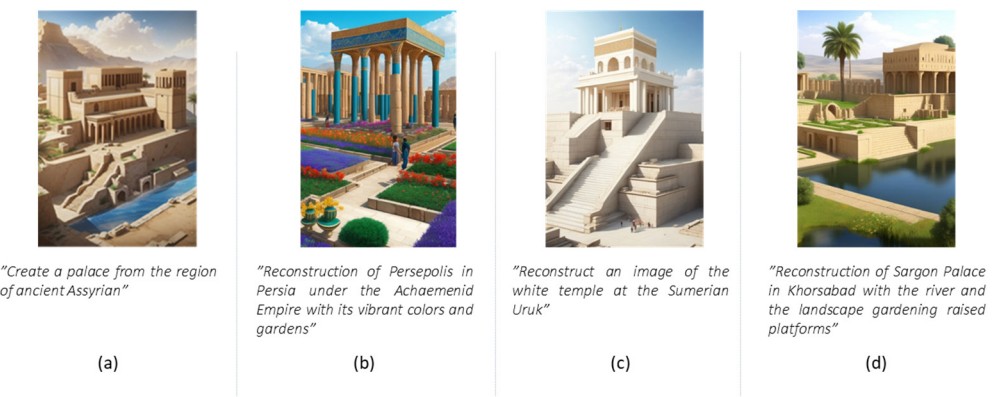

"Create a palace from the region of ancient Assyrian"

(a)

"Reconstruction of Persepolis in Persia under the Achaemenid Empire with its vibrant colors and gardens"

(b)

"Reconstruct an image of the white temple at the Sumerian Uruk"

(c)

"Reconstruction of Sargon Palace in Khorsabad with the river and the landscape gardening raised platforms"

(d)

**Figure 10.** In Workshop B, Task 4 re-envisioned different examples of lost cultural heritage from ancient Persian, Sumerian, and Assyrian civilizations.

4.2.2. Feedback from Students

The workshop had a multinational character, with representatives from the United Arab Emirates, Jordan, Palestine, Syria, Egypt, Iran, Kuwait, India, and Iraq. The ages were diverse, with students aged between eighteen and twenty-one, offering different viewpoints. Regarding knowledge of AI image generation, one participant had a shallow understanding (rating 1), four participants had moderate familiarity (rating 3), three were more-familiar participants (rating 4) and, lastly, three were deeply knowledgeable participants (rating 5).

Figure 11, showcasing the results of the questionnaire, focuses on the inventive aspects of AI image generators in educating on the "History of Architecture" (with a total of 11 participants) Several trends emerge. In comparative studies between historical layers (Task 1), most participants expressed a neutral stance on confidence, indicating a balanced outlook with five responses. At the same time, (3, $n = 11$) one strongly agreed, and two somewhat agreed. However, opinions on this challenging nature varied, with three strongly disagreeing and two somewhat agreeing. Regarding satisfaction, responses were more positive, with (4, $n = 11$) somewhat agreeing and three strongly agreeing. Moving on to designing modern buildings in historical styles (Task 2), a notable trend is the overall confidence and satisfaction among participants, with six who strongly agreed and five who somewhat agreed, indicating a collective belief in the feasibility of this application. Interestingly, opinions on the challenging aspect were more divided, with three strongly agreeing and one each for somewhat agreeing, neutral, and disagreeing. Regarding AI versus human intelligence (Task 3), the responses land towards confidence and satisfaction, with seven somewhat agreeing and three strongly agreeing. At the same time, challenging aspects received mixed responses, including three strongly disagreeing. Finally, virtual reconstruction of incomplete or destroyed architecture (Task 4) garnered diverse opinions, particularly on confidence, where (5, $n = 11$) expressed neutrality and four disagreed. Challenges were evident, with (5, $n = 11$) strongly agreeing, and satisfaction was still perceived positively, with five somewhat agreeing.

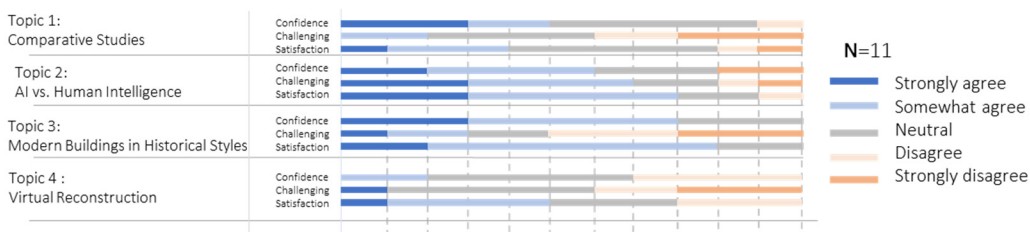

**Figure 11.** Results of the questionnaire (see Appendix B): each row shows an item scaled from strongly agree (blue) to strongly disagree (red), while each cell represents the answer of one participant.

After the online questionnaire, participants were asked about their interest in using this AI-based tool in future courses—their responses revealed a range of perspectives. No one expressed a complete lack of interest (rating 1). One participant showed limited interest, while two had a moderate interest (rating 3). The majority, comprising five individuals, exhibited a relatively high interest in using such a tool in future courses. The participants were very interested, providing the highest rating of 5. This rating reflects the diverse attitudes among participants regarding their willingness to incorporate this tool into their future educational experiences.

Participants shared diverse experiences and reflections on using AI to restore ancient structures visually. Some found AI's capacity impressive, while others called for refinement. Many applicants enjoyed this experience and expressed gratitude, despite the varying AI efficacy. Challenges were encountered in obtaining desired results, with suggestions for more detailed requests. Participants acknowledged the AI limitations in distinguishing historical epochs and architectural nuances, particularly for heavily damaged structures.

*4.3. Feedback from Educators*

During October and November 2023, five experts in architectural history education were interviewed. These included academics from various architectural departments within the UAE, Oman, Saudi Arabia, and Egypt. Interviewed experts generally agreed that "History of Architecture" courses would benefit from AI image generators as they would allow students more engaging opportunities due to their fast and expressive nature. Nonetheless, some concerns have been expressed about the reliability and authenticity of AI-produced visualizations from different historical eras, especially regarding less information. The range of perspectives expressed by the experts on the studied experiments is shown in Figure 12.

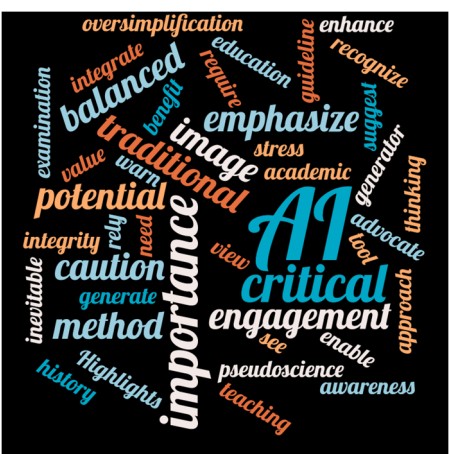

**Figure 12.** Word cloud showing the range of perspectives expressed by the experts on the studied experiments.

Firstly, educators were asked about the benefits of employing AI image generators as a pedagogical tool for teaching architectural history. Experts agreed that AI image generators can enhance the interactive teaching and learning environment by providing a wide range of visual representations of historic buildings and sites, enabling students to explore numerous buildings and better understand architecture evolution. Expert 1 argued that "*Like it or not! The use of AI image generators will soon be inevitable*". When used judiciously, AI image generators offer numerous benefits. They serve as a platform for students to test their knowledge, enhance their understanding of historically built environments, and can potentially become key content for blended learning initiatives. "*The certainty of using AI image generators in architectural education necessitates a proactive approach, urging early accommodation and proper utilization; using AI in teaching courses will not compromise learning outcomes but will instead make it more engaging and informative*" (Expert 2). Expert 4 did not oppose the proposed integration, but emphasized the importance of "*students forming their own opinions based on a comprehensive understanding of the underlying facts, AI image generators to be a significant opportunity for students to develop critical thinking skills within architectural history instead of the temptation of quick takeaway products*".

The second question of the interview urged the experts to share their considerations and concerns about integrating AI into the education of architectural history. Of particular interest were reflections on ensuring the accuracy and reliability of the AI-generated images, addressing potential challenges that may arise in this regard. Expert 1 warned against "*the rush transformation to AI image generators without appropriate preparations, using AI for image generation as a learning tool should commence after students have acquired a sound knowledge of architectural characteristics specific to historical periods*". Expert 2 pointed out the ongoing challenges of the controllability and interactivity of existing AI image generator tools: "*Future tools should be more interactive and adaptable, May be combining both image-to-image and text-to-image techniques interchangeably*". *Moreover*, Expert 3 also underscored the importance

of *safeguarding against possible student misuse,* "*Despite its technological prowess, the mechanism failed to capture the profound philosophical dimensions of human understanding. At the same time, it can vividly depict architectural elements such as domes and minarets in Islamic architecture; it remains incapable of encapsulating the intangible factors that contributed to the genesis of these marvels*" (Expert 4).

The third question of the interview focused on assessing the effectiveness of AI-generated images in facilitating student learning and engagement within architectural history. Expert 1 recommended empowering students with the capacity to guide AI with an architectural framework, enabling them to make detailed inquiries and assess the accuracy of generated images. While recognizing the effectiveness of AI-generated images in encouraging students to think critically and define questions for accurate results, Expert 1 cautioned against the complete replacement of traditional teaching methods. Expert 2 looked at how architectural visualization tools have developed historically, starting with the traditional manual methods in the pre-1980s, Computer-Aided Drawing (CAD) in the 1990s, and Building Information Modeling (BIM) in the 2010s, pointing out that "*artificial intelligence (AI) may become the new norm by the 2020s especially after the coronavirus pandemic Which shows the importance of providing general guidelines for "AI-enabled architectural education" while upholding academic integrity and addressing ethical issues such as plagiarism and copyright with specific policies specially designed for each courses to offer a complete and well-rounded approach, especially when dealing with cultural-related courses such as the history of architecture*". In a similar vein, Expert 1 expressed skepticism regarding some AI-generated images that had become widespread in recent months on social media, drawing parallels to "*an orientalist view of medieval historic architecture and cities*" in certain instances. Highlighting concerns, Expert 2 contended that "*while AI tools provide fast visualization, History is not solely about images, there should be a clear warning against the potential Disneyfication of architectural history if reliance on AI image generators is unchecked. The abstraction and dilution of the history of architecture discourse into visuals significantly undermine the impact and true purpose of teaching architectural history, which should delve into socio-economic and political domains that shape architectural products*". Consequently, Expert 2 recommended that "*AI-generated images should serve as an assisted tool rather than the sole source*".

The authenticity of the generated images is contingent upon the dataset's development by users rather than academics, leading to a risk of introducing speculative elements into the representation of historical architecture. To ensure the precision and reliability of depictions, Expert 5 emphasized "*a preferable approach involves showcasing genuine images of actual buildings. This task is best undertaken by academic experts who deeply understand historical and cultural contexts*".

Lastly, these experts were invited to share their perspectives on the future of AI image generators in architectural history education. "*No doubts, the interest towards AI generally is and will grow unprecedently. However, the need for accuracy verification of the produced images should depend on students' knowledge and critical thinking skills developed through conventional methods*" (Expert 1). Adding to that, Expert 4 characterized this tool as a means for exploring diverse scenarios, but contended that it did not achieve the status of genuine science. Instead, he suggested "*viewing it as a pseudoscience that needs a rigorous critical examination*". Furthermore, Expert 4 suggested developing "*specific AI image generator tools tailored for teaching the history of architecture which adopt a balanced approach where students acquire historical facts through conventional methods before integrating AI as an additional layer to enhance engagement and support understanding. This approach aims to mitigate the possible risks of oversimplifications and distortion in the educational process*". Similarly, Expert 5 pointed out, "*While these AI tools can engage students, fostering creativity and a sense of fun, the educational value must be balance with a critical awareness of potential inaccuracies. Like any other teaching tool, there should be more emphasis on developing critical reading when viewing historical representations*".

## 5. Discussions

### 5.1. Bloom's Taxonomy Applied to AI Image Generation for Educating on the HoA

As explained earlier in the related work section, the application of Bloom's taxonomy of educational learning objectives to current practices in the HoA (Figure 2) is observed to be basically focused on low-order thinking skills, with a notable gap in addressing higher-order cognitive abilities. The mentioned results might bridge this gap, as shown in the following:

- As evidenced in Task 4 (B.4 in Figure 13) in Workshop B, supporting traditional face-to-face debates and discussions with visual counterparts can open new spaces for thinking and methods for both educators and students, fostering a more substantial analysis of the studied subject or historical era. This exchange of ideas between verbal discussions and visual depictions can contribute to a more prosperous and dynamic exchange of architectural ideas.
- Workshop A, and Tasks 1 and 2 (B.1 and B.2 in Figure 13) from Workshop B, are possible additions to improve higher-order thinking skills associated with bridging the gap between analyzing and the creation processes, in a smooth transformational manner.
- The potential coupling of human thinking with AI was demonstrated by the third task of Workshop B (B.3 in Figure 13). This added another view to the "*Create*" section, in addition to the traditional design projects.

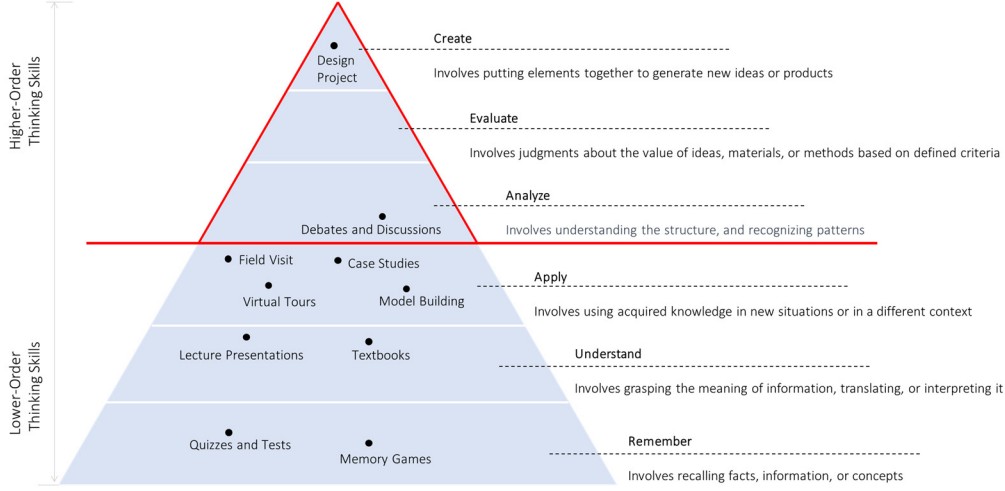

**Figure 13.** The incorporation of AI image generators aligned with the higher-order thinking skills of Bloom's taxonomy in the education of the HoA.

### 5.2. Reliability and Validity of Historical Content in Teaching the HoA

The issue of accuracy in educating on the HoA is undebatable, in order to ensure that the students have enough abilities to understand architectural elements, styles, and historical contexts. The images produced during the two workshops were mainly powerful in terms of visual appeal and color. However, some inaccurate content appeared, raising reliability concerns. These accuracies included exaggerations of particular architectural elements, inaccurate colors, and misrepresentations of geometrical outlines, such as in the case of the Ziggurats in Workshop A (Figure 3), which featured circular buildings instead of historically accurate rectangular ones.

The interviewed experts showed different views regarding inaccuracies. Expert 1 emphasized the urgency of the need for accuracy verification based on students' knowledge and critical thinking. Expert 4 went further, describing AI-generated images as a form of pseudo-science that needs to be rigorously examined. A good approach that can be adopted to tackle this issue is applying educational psychology concepts such as critical pedagogy [44], which encourages students and educators to question, analyze, and criticize

the presented information by gradually developing inquiry-based learning and analytical skills. Moreover, critical pedagogy can foster a culture of questioning and empowering the students to use the AI in a responsible manner. When coupled with traditional tools, this can enable students to actively seek out information, critically analyze the content, and further engage in discussions. Therefore, inaccuracies become a means of promoting continuous learning and adaptation. This research emphasizes the complex relations between critical pedagogy and human and AI understanding. With this in mind, a new chapter of human–computer interaction can be further studied.

### 5.3. Educating on the HoA for Non-Architects

Architectural drawings are specialized visual representations that convey complex spatial, symbolic, and technical aspects that may be unfamiliar to individuals without a background in architecture. This unfamiliarity appears from the variations in cognitive perspectives between specialists and non-specialists. For instance, the results of a previous study on the exploration of the façades of historical residential buildings regarding cognitive differences in aesthetics [45] showed significant variations in evaluating the studied architectural elements. Architects emphasize spatial aspects, while non-specialists focus more on the quality of the façade ornaments. Moreover, the perception of interior environments in a historic building was explored in [46], highlighting non-experts' tendency towards the architectural experience rather than its technical dimension.

Workshop A showcased that students majoring in diverse fields without prior architectural experience were able to visualize and comprehend architectural tasks to some extent. Moreover, those students added fresh perspectives on conventional architectural ideas and terms. It is also argued that AI-generated images can function as a bridge to build common ground and open up discussions between students and their educators, engagingly and inclusively, despite possible controversies and inaccuracies. However, this notion needs to be proved and further applied within similar courses in future research.

### 5.4. Capability of Comparing between Various Historic Layers

As the results of Task 2 in Workshop B may suggest, AI image generators can catch the distinctions between elements of different architectural styles, despite some inaccuracies that may concern the experts and educators about whether the students can recognize the variation without mixing the styles together.

According to the interviewed experts, while AI offers a promising direction for comparing historical layers, contextual understanding still needs to be improved. It necessitates a cautious review compared to the human touch. In this sense, approaching AI-generated interpretations with a critical mindset in the HoA courses is highly recommended. This collaborative approach ensures a more comprehensive and nuanced examination of historical architectural layers, combining the strengths of AI technology with the interpretative power of human historical expertise.

### 5.5. Hybrid Intelligence: Coupling AI and the Human Brain

The results of the third task in Workshop B highlighted the possibilities of combining human thinking with AI. According to the studied feedback, most of the students were able to critically recognize the differences and similarities between their own sketches and AI-produced images. The interviewed experts voiced concerns about the extent of the dependency of some students on AI tools instead of their intellectual and critical thinking; however, a more controlled approach to this human–AI collaboration can be beneficial in interpreting the design and historical aspects of the built environment.

### 5.6. Exploring the Alternative Histories of Unfinished or Damaged Architecture

AI image generators were able to offer picturesque depictions of cities, buildings, and elements that are currently lost, destroyed, or partially demolished, as observed in the last task of Workshop B. The students were allowed to experience these structures like

never before, creating an unprecedented historical narrative of their surroundings and contexts, especially when it came to inaccessible sites. Moreover, the ability to visualize how incomplete or damaged buildings might have looked under different historical periods introduces a range of possibilities for historical reconstruction, not only in the physical aspect but also in the virtual realm. For instance, Gamification can play a role in further exploring such issues, with an exploration of the spatiotemporal analysis of a particular building which can encourage educators and students to critically discuss what could have affected the building through its historical timeline. Regardless of their visually compelling images, a careful approach should be practiced to reduce the possible biases and inaccuracies resulting from AI algorithms. Hence, treating the AI-produced images as supplementary tools for catalyzing and deepening understanding instead of replacing traditional tools is highly recommended.

### 5.7. Technicalities and Usability

The ease of use and accessibility of the user interface was evident in both workshops, especially to the students of Workshop A, who came from diverse technical backgrounds. However, the extent to which the educator or the student can modify the parameters of the resultant image is still limited.

Our discussions with experts agreed on the ease of use and access, and further emphasized the need for new tools specifically tailored for HoA courses. Undertaking this step is expected to pose challenges, as it requires substantial time, funds, unbiased sources, and technical skills.

## 6. Conclusions, Limitations, and Future Work

This research has demonstrated the possibilities of integrating AI image generators within HoA courses. The workshops enabled the students to experience various ideas and contexts based on their understanding. The results of Workshops A and B and their feedback showed several points that need further discussion concerning the idea of coupling AI power with human thinking, the strengths, weaknesses, reliability, and the accuracy of AI-generated images. The suggested integration method for AI image generators into HoA courses holds significant promise. Nevertheless, we strongly advocate for viewing this method as a supplement rather than a substitution for the established traditional approaches.

The research hypotheses show promise to a large extent, despite the presence of several limitations. Firstly, the workshops were limited in location and period as they were held at the same university and lasted only two hours each as one-day workshops. Future workshops can be extended to one week or more, to allow more time for discussions and experimentation with other historical contexts. Moreover, the workshops' content focused on some lessons as a pilot, not the whole course syllabus. Future workshops can be more comprehensive, with more architectural style, terms, and ideas to be tackled. On the other hand, Leonardo AI was primarily used to avoid cost and accessibility issues. It would be more beneficial to reconduct the same workshop with other AI image generators, such as Stable Diffusion or Midjourney, which would enrich the research discussion through a comparative analysis.

The significance of this research lies in the introduction of a new tool to traditional architectural history education, and therefore it opens up the road for more future research directions, while knowing that AI image generation is a technology in its early stage. Firstly, more research is needed to validate the accuracy of AI-generated images, mainly depending on the students' knowledge and critical thinking. Hence, a rational balance should be reached between the intellectual strength of human thinking and the technological power of AI. Secondly, a framework of guidelines should be adopted to reduce the possible misuse of the tool and to be more inclusive in terms of cultural and linguistic variations. Future research might also consider using alternative pedagogical approaches or theoretical frameworks relevant to architectural history education, demonstrating a

more comprehensive understanding of the possible relationship between pedagogy and architectural history with AI-generated images. Finally, specially tailored AI tools can be designed for HoA courses to increase student control and interactivity while avoiding over-simplifying architectural styles.

**Author Contributions:** Conceptualization, M.W.F., A.B.N. and E.N.; methodology, M.W.F. and E.N.; software, M.W.F., A.B.N. and E.N.; validation, M.W.F., A.B.N. and E.N.; formal analysis, M.W.F. and E.N.; investigation, E.N. and M.W.F.; resources, M.W.F.; data curation, M.W.F.; writing—original draft preparation, M.W.F.; writing—review and editing, A.B.N. and E.N.; visualization, M.W.F.; supervision, A.B.N. and E.N. All authors have read and agreed to the published version of the manuscript.

**Funding:** The first author is grateful for ICCROM-Sharjah for the Master Scholarship 2022–2024.

**Data Availability Statement:** All data mentioned in the paper are available through the corresponding author.

**Acknowledgments:** The authors are grateful to the BSc and MSc students who participated in the workshops for their contribution and feedback. The authors would like also to thank all educators of History of Architecture who participated in the interview to share their observations and experiences.

**Conflicts of Interest:** The authors declare no conflicts of interest.

## Appendix A. Surveys for Students to Measure the Effectiveness of Workshop A

The development of a general framework for using machine learning techniques in teaching architectural history.

This workshop focuses on tools that convert text into images to enhance educational experiences in architectural history. The objectives include:

- Enhancing understanding of architectural history through realistic images generated by artificial intelligence techniques and image creation.
- Introducing students to the concept of text-to-image tools in the context of architectural education.

**Required \***
**Section 1: Workshop Overview \***

- Age: ..........................................
- Country: .........................................
- Educational Background (Bachelor's Degree): ....................................................

**Section 2: Workshop Content (Rate from 1 to 5) \***

|  |  | Strongly Agree | Agree | Neutral | Disagree | Strongly Disagree |
|---|---|---|---|---|---|---|
| [1] | How clear was the topic of using artificial intelligence in teaching architectural heritage to you? |  |  |  |  |  |
| [2] | Do you think the workshop adequately covered the topic? |  |  |  |  |  |
| [3] | How much do you appreciate the proposed framework for using artificial intelligence in teaching architectural heritage? |  |  |  |  |  |

**Section 3: Deconstructing "Architectural History" (Rate from 1 to 5) \***

| | Strongly Agree | Agree | Neutral | Disagree | Strongly Disagree |
|---|---|---|---|---|---|
| [1] Please assess your belief in how the proposed tool will help explain historical background concepts in architectural history courses. | | | | | |
| [2] Please assess your belief in how the proposed tool will help explain the function concepts of buildings in architectural history courses. | | | | | |
| [3] Please assess your belief in how the proposed tool will help explain aesthetic and visual features in architectural history courses. | | | | | |
| [4] Please assess your belief in how the proposed tool will help explain realistic aspects in architectural history courses. | | | | | |
| [5] Please assess your belief in how the proposed tool will help explain non-material (intangible) aspects in architectural history courses. | | | | | |

### Section 4: Workshop Impact*

- Do you intend to implement the proposed smart framework in your own teaching or educational activities? If not, please specify your reasons. ...................................................
- Would you recommend the workshop and the smart framework to other teachers or colleagues? ...............................................................
- Do you have any specific suggestions to improve the workshop or the smart framework? ...............................................................

### Section 5: Open Questions*

- How can machine learning applications be experimentally applied to enhance education in the field of architectural heritage in the Middle East and North Africa region? ...............................................................
- How can machine learning applications improve students' understanding of architectural heritage concepts? ...............................................................
- What is the impact of machine learning applications on teachers' and students' engagement and interaction towards education in the field of architectural heritage? ...............................................................

### Appendix B. Surveys for Students to Measure the Effectiveness of Workshop B
### * Required

This workshop focuses on text-to-image conversion tools to enhance educational experiences in architectural history. The objectives include:

- Enhancing understanding of architectural history through realistic images generated by artificial intelligence techniques.
- Introducing students to the concept of text-to-image tools in the context of architectural education.

### General Information

- Country of origin: . . . . . . . . . . . . . . . . . . . . . . . . . . . . . . . . . . . . . . . . . . . . . . . . . . . . . . . .

- Age: . . . . . . . . . . . . . . . . . . . . . . . . . . . . . . . . . . . . . . . . . . . . . . . . . . . .
- Rate your familiarity with AI image generators (from 1 to 5): . . . . . . . . . . . . . . . .
- Years of studying architecture: . . . . . . . . . . . . . . . . . . .

**Mini Experiment 1: Virtual Reconstruction of Unfinished or Damaged Architecture:**

| | | Strongly Agree | Agree | Neutral | Disagree | Strongly Disagree |
|---|---|---|---|---|---|---|
| [1] | How confident do you feel about the ability of AI to visually reconstruct an unfinished or damaged ancient building on a scale of 1 to 5, with 1 being not confident at all and 5 being very confident? | | | | | |
| [2] | Did you find it challenging? | | | | | |
| [3] | How satisfied are you with the final products you created on a scale of 1 to 5, with 1 being very dissatisfied and 5 being very satisfied? | | | | | |
| [4] | Please describe your experience and thoughts during the process of visually reconstructing the ancient building. | | | | | |

**Mini Experiment 2: Architectural Evolution Visualization:**

| | | Strongly Agree | Agree | Neutral | Disagree | Strongly Disagree |
|---|---|---|---|---|---|---|
| [1] | How satisfied are you with the final products you created on a scale of 1 to 5, with 1 being very dissatisfied and 5 being very satisfied? | | | | | |
| [2] | Did you find it challenging? | | | | | |
| [3] | How confident do you feel about the ability of AI to visually show the evolution of a particular typology or architectural era on a scale of 1 to 5, with 1 being not confident at all and 5 being very confident? | | | | | |
| [4] | Share your thoughts on the experience of creating visualizations of architectural evolution. | | | | | |

**Mini Experiment 3: Comparative Studies between Historic Layers:**

| | | Strongly Agree | Agree | Neutral | Disagree | Strongly Disagree |
|---|---|---|---|---|---|---|
| [1] | How satisfied are you with the final design you created on a scale of 1 to 5, with 1 being very dissatisfied and 5 being very satisfied? | | | | | |
| [2] | Did you find it challenging to design a building that combines features of two different historical architectural styles (e.g., ancient Persian and Egyptian)? | | | | | |
| [3] | How confident do you feel about the ability of AI to visually combine architectural styles on a scale of 1 to 5, with 1 being not confident at all and 5 being very confident? | | | | | |
| [4] | What were the main challenges or insights you gained from this experiment? | | | | | |

**Mini Experiment 4: Designing Modern Buildings in Historical Styles:**

| | | Strongly Agree | Agree | Neutral | Disagree | Strongly Disagree |
|---|---|---|---|---|---|---|
| [1] | How confident do you feel about the ability of AI to design modern buildings in historical styles on a scale of 1 to 5, with 1 being not confident at all and 5 being very confident? | | | | | |
| [2] | Did you find it challenging? | | | | | |
| [3] | How satisfied are you with the final design you created on a scale of 1 to 5, with 1 being very dissatisfied and 5 being very satisfied? | | | | | |
| [4] | Share your thoughts on the experience of blending modern and historical architectural elements. | | | | | |

**Mini Experiment 5: AI vs. Human Intelligence:**

|  |  | Strongly Agree | Agree | Neutral | Disagree | Strongly Disagree |
|---|---|---|---|---|---|---|
| [1] | After using an AI image generator, please rate the similarity between your sketch and the AI-generated image on a scale of 1 to 5, with 1 being very dissimilar and 5 being very similar. |  |  |  |  |  |
| [2] | Did you find it challenging? |  |  |  |  |  |
| [3] | How satisfied are you with the final design you created on a scale of 1 to 5, with 1 being very dissatisfied and 5 being very satisfied? |  |  |  |  |  |
| [4] | What did you find most interesting or surprising about the comparison between your sketch and the AI-generated image? |  |  |  |  |  |

- Overall, would you like to use such a tool in future courses? .........................
- Which one of the mini experiments did you find more engaging? Why? ...............

**Notes**

1    https://www.midjourney.com/home?callbackUrl=/explore (accessed on 15 December 2023)
2    https://openai.com/dall-e-2 (accessed on 17 December 2023)
3    https://leonardo.ai/ (accessed on 12 December 2023)
4    https://stablediffusionweb.com/ (accessed on 10 December 2023)

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
