# Peer review of "Exploring the Potentials of Artificial Intelligence Image Generators for Educating the History of Architecture"

_heritage, doi:10.3390/heritage7030081_

Round 1

Reviewer 1 Report

Comments and Suggestions for Authors

A very interesting and important article, but with a fundamental flaw in one of its basic assumptions. In my opinion, the methods proposed by the authors are not intended to teach the history of architecture, but to create certain new values based on historical knowledge. Learning the history of architecture is based on knowledge of facts, context and the ability to interpret them, and not on the creation of new, often abstract forms, which are strongly dependent on the student's knowledge and his ability to introduce it to AI. The only element proposed by the authors that, in my opinion, can support learning the history of architecture - is the visualization of objects in their historical surroundings - which can greatly support students' understanding of the context and urban significance. However, this task also requires introducing scientific data into AI consistent with research results and current knowledge. At the same time, the methods proposed by the authors seem perfect for transferring knowledge and skills related to heritage protection and conservation design. The above-mentioned analyzes also seem to confirm this interpretation. In turn, the quoted statements of experts largely confirm my doubts. Therefore, perhaps a change should be made in the title and basic assumption of the article - because the didactic possibilities of using AI have been indicated and justified by the authors, so its basic content is beyond doubt. I also have a few specific comments on the article, listed below:

Fig. 1.: The drawing is very legible. The only thing that causes some discomfort is the indescribable gray band between the ZPD area and the outer ring. As I understand it, this is a graphic procedure intended to increase the readability of the drawing. At the same time, however, it may suggest the presence of another "substantive" layer here. Perhaps it is worth rethinking this graphic to make it more clear?

line 88: Please use full names the first time they are used in the text (abbreviations are absolutely sufficient for subsequent uses). I know that abbreviations such as VR or AR have already become widely used, but I think that this will improve the readability of the text and make it possible to refer to it even when these abbreviations are no longer so obviously recognizable...

line 105: ...what you did in line 105 ;-)

line 184: The word "castle" means a defensive facility. Meanwhile, Versailles is a former royal residence complex without any defensive features - so the word "palace" would be much better here.

Figures 3, 4, 5, 7, 8: Are the captions under the figures an exact or abbreviated transcript of the prompt given to the AI, or do they merely describe the purpose of the task? Please elaborate on this.

Fig. 4.: The name Çatalhöyük is incorrectly spelled in the caption.

Fig. 6.: Please provide the exact content of the question answered by respondents.

Fig. 11., line 475 - 509: Without knowing the questions answered by the respondents, this part of the article is completely incomprehensible.

Reviewer 2 Report

Comments and Suggestions for Authors

The paper addresses a strictly current issue whose discussion is not only necessary but urgent. However, the topic requires an in-depth study of the disciplinary boundaries in the interactions between traditional teaching and learning and the use of digital tools, which the authors have recalled but have not analyzed in depth (for example the boundaries between virtual and augmented reality and between these and artificial intelligence). Paragraph 2 should be further developed precisely for the aforementioned reasons.

The paper also presents an ambiguity in moving between the specific domain of architectural history and the pedagogical analyzes relating to teaching and learning styles but overall both these domains are analyzed superficially. In particular, reference is made to a precise model, Bloom's Taxonomy, which places historical knowledge as the foundation for planning. However, there are many alternative approaches and the scientific production of architectural historians, from Tafuri to Wittkower, from Vidler to Kaufmann or Leach, amply demonstrates the autonomy of the history of architecture from the design phase; the authors ignore this aspect or should at least clarify why they rely only on Bloom's model.

Paragraph 3 adequately illustrates the two case studies, even if the authors sincerely declare that the limited times of the workshops conditioned the work and restricted the research to well-defined territories and periods (this, however, could also be an advantage of the research).

However, there is no final critical evaluation by the authors. Specifically, the results of the two workshops processed information relating to very important buildings and modeled this information in the output results. But one of the main tasks of the history of architecture is to acquire knowledge about that specific architecture in its singularity, not about how it could have possibly been

Reviewer 3 Report

Comments and Suggestions for Authors

0: First, I would like to commend the idea of your research. Below, I have outlined what I think would improve your text.

20: Specify that this is a subject called History of Architecture. In case it is a field, write it in lowercase and add a qualifier that it is a field (e.g., field of history of architecture).

28-33: Rephrase these findings to be presented briefly and clearly. In this part of the abstract, it is necessary to briefly mention the results you have achieved. If you do this in an interesting way, you can expect someone to read your paper because they are interested in how you arrived at these results.

35: Add Leonardo Ai

39. This sentence ambiguously emphasizes the significance of architectural history for future architects. I suggest taking a different approach. For example: The job of an architect is inconceivable without considering the broader picture of a task. As architecture is a social discipline, the development of architecture is linked to the development of society, i.e., its history. The history of architecture is one of the subjects taught in architecture schools, so that future architects can engage in various tasks, which are often related to the field of the history of society, i.e., the history and theory of architecture and art.

49: Replace "Ai" with the full name, put the abbreviation in parentheses, and use it accordingly in the rest of the text. Also, tie that term to footnotes or references at the end. This applies to the entire text.

51: You have mentioned the name of the course. I think it's better to state that it is only about the field of history of architecture, and later specify the exact name of the subject, which year of study it is taken in, how, and at which university.

60: Clearly and concisely state your research hypothesis. At the end, simply state whether you have positively or negatively answered this assumption.

84. In addition to the reference dealing with Bloom's Taxonomy, link this term to a new reference or link. It is also useful to briefly describe the idea of this taxonomy and why it is important for your course (e.g., the syllabus writing instructions at your university require you to rely on knowledge, skills, and competencies).

88: Provide full names and links for VR and AR.

101: Image 2 should come immediately after its citation in the text. Assume that later, during the production phase of the paper, you will arrange the text and images so that their layout corresponds to page breaks.

107-113: This is not accurate. If you have photogrammetry scans, they will provide a completely different basis for research. In this part, you imply that you will use Ai tools to achieve new results. Later, you present these results, discuss, and draw conclusions. Only then could you say that these tools are necessary.

122: Especially for critical thinking in this part of the research presentation, you can imply that students will develop this way of thinking by using AI tools. My personal opinion is quite the opposite because when I read your paper, I was not convinced that you had proved it. Nevertheless, I support the publication of this paper, but you would have to correct such claims.

127: Here you have explained the significance of architectural history well. Again, until you present a specific course, use the term "field" instead of the subject name (126).

138: Briefly describe the structure of your entire course by areas. Provide the name of the university, faculty, and relevant links.

150: Instead of "Researchers in (15)..." use: In the study dealing with (state what) (15), the authors demonstrate...

198: Stable Diffusion

169: In this chapter, briefly describe how the described tools work. It's actually about machine learning, and it's clear that AI is used to update the paper's titles (to attract focus). However, you need to show (refer to works in this field) how these tools actually work. Then explain why you will use the AI term instead of ML.

224: You chose Leonardo because of the price... that's okay and should be stated.

228: Add that these are one-day workshops.

227: Specify the exact names of the organizers (institution) and that the authors of this paper are also the authors of the workshops. If the list of authors is longer, this chapter should list all authors or reference where it is described. Also, how did you recruit students? For example, the authors were also selectors because 150 students applied.

266: Based on what did students create prompts?

278-279: This is not clear. You encouraged students to explore, using the current popularity of Ai tools, according to your goal to explore whether these tools can develop creativity. In my opinion, by using these tools, you made your field more interesting, and the development of critical thinking should perhaps be shown in another paper. I believe this was one of the limitations of this research.

271: How many students participated in workshop B?

281: Here you nicely stated that the goal is to explore whether Ai can incorporate the idea of tradition into contemporary design. Here you actually moved away from the topic of history and into architectural design. I think you should say that you wanted to connect the importance of history and contemporary design to check how Ai tools can interpret heritage ideas.

284. Are you sure it's about creativity? From the drawings, one cannot conclude that they are creative works (I'm sorry). You should conclude that AI has developed 3D visuals from student sketches. The field of architectural history is certainly not about assessing new architecture but about presenting and analyzing certain phenomena in the past. However, you can evaluate (because you are competent) how successful AI is in creating a representation given as an idea in the form of sketches.

335: State whether the entire prompt or a part of it is given below the visuals. I believe you should mention one process from start to finish. List all prompts. Then say that a similar process was conducted for b, c, and d.

369: Provide chatbot links and references.

411: At the end of this chapter, you should also present your evaluation of visuals at the workshop.

510: Like with students, describe the impressions of experts and summarize them in the form of graphs. I think it's not necessary to quote their words.

601: In the Discussion chapter, you need to connect your research with the literature. In this part, you evaluate yourself and show whether you are in line with certain opinions published. 

604: State HoA as the abbreviation for your course at the beginning of the research.

626: This is a good interpretation of the results.

646: Here you have applied critical thinking. Emphasize and add some more explanation of the significance of your research.

682: This is very important. It represents an excellent outcome of the research. You can also add in the introduction that you assume this will happen. Thus, using the scientific method, you demonstrate that, for example, 1+1=2.

712: Instead of "we demonstrated" use "This research has shown...". Sentences with "we..." should be avoided in scientific writing. Use the impersonal form.

718: State precisely whether your hypotheses from the beginning have been shown to be correct or not.

Round 2

Reviewer 1 Report

Comments and Suggestions for Authors

I'm glad to see that most of my comments have been incorporated into the revised paper. I understand your arguments (although I don't agree with all of them). However, I kindly request strengthening the information in the introduction and conclusion that the proposed method complements standard methods of studying architectural history but cannot entirely replace them.

Author Response

Thank you for your constructive feedback and for your suggestion, We have added the following accordingly:

•    In the 'Introduction' Section (77-82):
Considering this, it is vital to note that while integrating AI into architectural education, particularly HoA, may be beneficial in certain aspects, it may also bring challenges. It should not be intended to entirely replace existing teaching methods. Therefore, edu-cators are tasked with preparing students for a future where AI plays a central role in architectural practices, emphasizing the importance of human insight and creativity in architectural education [10].

•    In the first paragraph of the 'Conclusion' Section (807-810):
The suggested integration method of AI image generators into HoA courses holds significant promise. Nevertheless, we strongly advocate for viewing this method as a supplement rather than a substitution for the established traditional approaches.

Reviewer 2 Report

Comments and Suggestions for Authors

No further comments.

Author Response

Thank you for the constructive feedback on our article!